# Farmers' Perception on Contract Farming in the Post-COVID Era: Empirical Study in Greece

**George Malindretos \*, Eleni Sardianou**  **and Maria Briana**

Department of Economics and Sustainable Development, Harokopio University of Athens, 70 El. Venizelou Str., 176 76 Kallithea, Greece; esardianou@hua.gr (E.S.); mbriana@hua.gr (M.B.)
\* Correspondence: gmal@hua.gr

**Abstract:** Contract farming (CF) as a sustainable practice has expanded rapidly, bringing numerous benefits to both the agribusiness industry and farmers, as well as the broader economy; CF is also considered a vehicle to tackle the challenges in sustainable development due to the serious effects of the COVID-19 pandemic. However, little attention has been paid to CF schemes in Greece. To address this evidence gap, the current study aimed to outline the socioeconomic profile of Greek farmers and how it is related to their perspective on CF in the post-COVID era. Primary data were collected in the agricultural area of central Greece, Thessaly, with a structured questionnaire containing three sections: demographics, awareness of the CF concept, and farmers' perceptions toward CF. The results indicate that 60.7% of the respondents are unaware of CF. Demographic and economic variables such as gender, years of experience, and income impact Greek farmers' attitude toward CF. In addition, factor analysis results reveal that economic benefits and social, technological, and environmental challenges and risks are associated with CF. We suggest that researchers and policymakers interested in the development of CF in Greece should consider the heterogeneity of the agricultural workforce for the development and successful implementation of policies related to CF. Educational programs towards increasing farmers' awareness and sufficient understanding of the practical issues of CF are also required.

**Keywords:** contract farming; socioeconomic characteristics; perception; sustainability; Greece

## 1. Introduction

Contract farming (CF) has been established worldwide as a practice that binds producers of agricultural products and firms, or even the state, within mutually beneficial schemes and contractual obligations. It has been defined as "an agreement between farmers and processing and/or marketing firms for the production and supply of agricultural products under forward agreements, frequently at predetermined prices" [1]. At its core, it is a practice to manage uncertainty and risk [2] for all participants. A vertical relationship links firms to producers of agricultural products [3], creating economies of scale and increasing both competitiveness and share in foreign markets [4]. Essentially, CF is described as a pre-agreement between two parties on four axes, namely, price, quality, quantity, and time allocated for the procurement of products [5]. It is about an institutional arrangement among a buyer who delegates the production of an agricultural commodity to a producer, a farmer, or farmers' cooperative [6]. Additionally, CF has been described as a cultural and historical phenomenon [7]. Successful CF schemes in practice require strong commitment [8], though a culture of mistrust may also be present and hamper efforts [9].

Global market presence requires a greater integration of agrifood value chains [10]. As a scheme, CF is popular in both developing and developed countries. Three areas of the world are considered to have a mostly unexplored potential in agriculture, Southeast Asia, South America, and Sub-Saharan Africa, though CF is not limited to them [11,12]. Legally speaking, CF is a diverse field [7]. There is a variety of possible types of CF contracts

depending on the product, which may include formal and informal contracts [13] and procurement, partial, or total contracts [5,8]. Five schemes, i.e., the centralized, the nucleus estate, the multipartite, the informal developer, and the intermediary(tripartite), are the most commonly used models [14], but they are not mutually exclusive [5].

Klonaris [4] argues that CF "reverses the most common production model that first produces and then looks for potential buyers". Its popularity is attributed to two reasons [15]: reduced risk and reduced transaction costs. Such risks are considered to be problems in market specifications, disincentives of all types as well as lack of coordination [2]. On the contrary, Adams et al. [16] argue that "contract farming changes rural agrarian relations, transforms local family institutions by carefully selecting a few household members with influence into the scheme and selectively dispossessing the poor community members". Problems related to CF such as biased terms, delayed or unfair payments, economic and social differentiation, high input costs due to a constant demand for high quality, high credit ratings, and land concentration are also examined in the literature [10,17–19].

CF is not unknown in Greece, although only recently it has taken a more organized form [14], though still focusing on traditional corps and processed products (e.g., tomato sauce). However, the number of farmers in CF schemes is relatively low; Greek farmers have been described as reserved, while banks are not particularly engaged in such practices. Research has pointed to a number of structural shortcomings of the Greek agricultural sector, like family ownership, for low numbers of CF schemes [20]. This trend may be reversing as following the EU's Common Agricultural Policy (CAP), the Greek rural economy has moved from the family farm system to CF [21].

However, there are still three crucial points for the Greek agricultural sector. The first one concerns the 2008 economic crisis that heavily affected the national and regional economies of Greece [22] and has "frozen" the activities of agricultural cooperatives all over the country [23], causing long-term effects on its welfare. The second relates to the COVID-19 pandemic that has also impacted the general population and, in particular, the agricultural workforce [24]. Since agriculture is the only provider of food inputs [25], the pandemic has undoubtedly created instability in the food sector along with global supply chain disruptions [26]. The last one is linked to the sustainability of the agricultural sector. Sustainable development involves economic activities that satisfy current needs without jeopardizing meeting future needs [27]. The rise and spread of the COVID-19 pandemic emphasized the adoption of sustainable farming techniques and agricultural practices to ensure food security [28]. Under this unstable economic environment for Greece, the role of agriculture as a sector of unique characteristics, as Gardner suggests [29], is redefined by embracing a collaborative culture. CF fits into this framework.

Given that the agricultural sector in Greece has always been a reference point for both economic and social life, the primary purpose of the study is to assess the perceptions of Greek farmers concerning CF. In this context, it attempts to explore how the profile of the farmers is related to CF awareness, how they perceive the CF concept, and how they assess the potential benefits, challenges, and risks arising from participation in a CF scheme.

This study contributes significantly to the emergence of CF as a useful tool for sustainable agriculture in the post-COVID era, providing a valuable contribution to an under explored research area through meaningful guidance for both raising awareness of CF and increasing the involvement of all stakeholders in Greece.

## 2. Literature Review

### 2.1. Who Joins a CF Scheme?

Many factors influence the decision of farmers to join a CF scheme. It has been noted that the diversity of possible schemes is so large that it is more convenient to focus on the specifics of each case [5] rather than generalizing. Demographics of farmers are of significant importance, though there is no consensus on their relative significance [15]. For instance, age, sex, and education level of farmers may be relevant; farm size and generally property rights [30], experience of the farmer, and exposure to risk and/or credit

are also mentioned. Kumar et al. [31] note that participation in CF is influenced by farm size and the main occupation of the cultivators. Farm size, smallholder's age, education, and participation in farm groups are also indicated by Simmons et al. [32] in their analysis of contract farming. According to Nsimbila [33], adopting CF is affected by gender, age, and experience in cultivation among other factors.On the other hand, education, farm size, and extension are factors that swayed the farmers' decision to engage in CF [34]. Finally, geography and local government policies are relevant [15,35]. Localized studies have shown that women, older farmers, and less educated individuals may be more unwilling to participate [36]. Of particular interest is the fact that powerful social norms restrict women's participation [37], thereby making them invisible farmers [38]. In any case, it should be noted that CF emphasizes on individual producers than cooperative groups [9].

Last but not least, the state may also engage in CF schemes. This is a common practice in China, where it may even be mandatory [39], and other countries [5,40,41]. In general, government policies, whether specific to CF or for agriculture in general, are important factors when studying such schemes [42], but we should distinguish CF from state partnership schemes as states in contrast with multinational companies may have different goals in production and management, therefore, targeting different groups of farmers [43]. Indirect influence of the state or stimulation of CF schemes could appear in the form of tax breaks and other incentives [44].

### 2.2. Why Join a CF Scheme?

Much has been discussed on the positive and negative effects of CF. Contracts in agriculture serve as a coordination device, provide incentives and penalties to motivate performance, and clarify the allocation of financial risk [45]. The literature notes that CF is the only way for farmers to access higher firms [15] or for smaller farms to overcome barriers to market [12]. Contract farmers may enjoy a number of guaranteed results, such as access to technological advances and a guaranteed supply of source materials all year long [46]. CF reduces the "hungry season", the part of the year where production of fresh products may be scarce by enhancing food security for a farmer [47]. In general, CF may contribute to farmers' welfare [48,49].

The willingness of producers to engage in a cooperative of some sort is associated with several advantages such as safer transactions, although not all producers seem to appreciate all advantages in the same way [50]. Competition is still possible in the scheme of CF in a particular area. Masakure and Henson [51] identify four CF motivations, namely, market uncertainty, indirect benefits (e.g., knowledge acquisition), income benefits, and intangible benefits (e.g., self-satisfaction). Moreover, the type of products under contract may be a factor that indirectly influences farmers that engage in a scheme. The scarcity of the product, its quality, local production and consumption, whether it is specialty crops, easiness of transportation to markets, requirements for freshness, and potential perishability should all be considered; for example, processing chain for tomatoes [8] is different than that of poultry and eggs [52]. Family-based labor or engagement in pre-existing cooperation schemes between farmers may be yet another factor for engagement in CF [34].

Several external reasons, like war and other major societal events, may also contribute to this decision [7]. Climate change is a factor that influences CF in multiple ways [12]. Technology, when it comes to processing, also affects engagement in CF. Finally, the interplay between farmers in CF and independents when they share the same lands is another aspect for consideration [7].

### 2.3. How a CF Scheme Is Perceived?

CF is often regarded to be a vehicle towards the transition of traditional agrarian communities towards modern agriculture in terms of technologies and procurement [30]. In particular, CF seems to have a positive impact on the technological and economic efficiencies of producers [53], especially those who rely on certain conditions for their products (e.g., rice producers). Positive results may vary from place to place and are not

limited strictly to agriculture. In Africa, for example, CF is considered as a game changer due to the potential of disrupting existing power relations [3]. Tuyen et al. [19] rank the perceived advantages of CF by the farmers as follows: (1) guaranteed price and reduced market fluctuations; (2) assured markets and possible access to new markets; (3) input and service provision; (4) access to credit;(5) access to inputs and services; (6) stable or increased income; (7) reliable supply of inputs and access to credit; (8) better product quality; (9) reduced pre- and post-harvest losses; (10) introduction to new techniques, new varieties, and practices; (11) improvement in farmers' skills and knowledge;and (12) access to advanced/appropriate technologies.

On the other hand, contracts in CF are thought to be biased against the farmers [8] or perhaps replicating negative aspects of colonialism [43]. Negative aspects also stress that farmers in CF schemes may only have an illusion of control for the farmers over the production [9]. Existing power structure disruption (i.e., preceding the local establishment of CF) may also have negative results [3] with firms exploiting farmers [54]. Economy-related drawbacks include, in case of the structure of markets, the creation of monopsonies and other market asymmetries. Additionally, farmers that focus on particular products risk deskilling [46]. Depending on the product, CF schemes may pose environmental risks [46], for instance, the overexploitation of natural resources like soil or water [8], especially for the production of non-edible or non-locally consumed products. Pesticides are often overused with all their negative effects [12]. Rout et al. [55] present the most important constraints experienced by farmers, namely, delay in payment of produce, the lack of credit for crop production, scarcity of water for irrigation, difficulty in meeting quality requirements, and lower prices of crop produce. The ranked order of farmers' perceived disadvantages of CF according to Tuyen et al. [19] is the following: (1) possible late purchase and input delivery, and delays in payments; (2) reduction in the household's freedom or loss of flexibility in making decisions; (3) manipulation of agreed quotas and quality specifications; (4) possible high price of inputs; (5) monopoly exploitation; (6) unequal bargaining power between farmers and contractors; (7) may buy less of the product than the pre-agreed quantities or be rejected for not meeting required standards; (8) possible greater environmental risks; and (9) risks of indebtedness from loans and excessive advances. As a result, farmers may commit contract breaches [56].

### 2.4. Contract Farming and Sustainability

CF is the backbone of modern agrifood value chains [39]. However, the interplay of CF and sustainability has not been wellresearched [30,57]. Heavy reliance on chemicals, like pesticides, and also reliance on limited resources, like water (especially for crops that require a lot), have been indicated among other practices as unsustainable practices for agriculture. CF and its demands for guaranteed output may be a burden for the environment and sustainability goals. Undoubtedly, promoting sustainable and balanced development across all rural areas is what matters. CF can be adapted to both comply with contract rules for production as well as SDG rules [30] and, thus, become an agent to spread good practices of sustainability within a local community of farmers. Therefore, farmers may both engage in CF schemes in order to meet sustainability rules [58], but also learn through CF engagement good practices that can ensure a long-term positive impact. In any case, sustainable CF is evaluated on case-by-case basis [59].

## 3. Methodological Framework

The target population of the current study consisted of farmers from Thessaly, a Greek region characterized as the most "agricultural productive" region in the country [60]. In 2022, about 17% of the workforce in Thessaly was employed in the primary sector of the economy, representing about 10% of the primary sector workers in Greece [61]. Additionally, Thessaly has more than 10% of the total cultivated land in Greece [61,62]. The survey took place during the first trimester of 2023. The surveyed farmers sampled were randomly selected by the researchers. The sampling frame used for the present study was as follows:

(1) researchers obtained the list of the population of all the farmers who were members of Cooperative Farmers of Thessaly, (2) a number was assigned to each farmer (numbers in the list were arranged so that each digit has no predictable relationship to the digits that preceded it or to the digits that followed. Thus, the digits were arranged randomly), (3) a random number generator was selected so as to choose the sampled farmer, and (4) the farmer corresponding to the selected number was included to the initial sample. A farmer could be selected only once. Thus, a random sampling without replacement was performed. A pre-testing process was also performed with input of two academics and four farmers on phrasing of questions in terms of clarity.During the surveyed period, the total number of members of Cooperative Farmers of Thessaly was 250 farmers. Surprisingly, the response rate was high, reaching almost 75%, and the survey resulted in a total data set of 150 farmers. Given the purpose of this study, farmers were interviewed at their businesses. To increase the probability of participation in the survey, the questionnaires were kept anonymous.

The questionnaire was a structured questionnaire, which included closed-ended questions and, in some cases, included a seven-point Likert scale. Given the limited number of empirical studies, the questions were formed not only by taking into account relevant previous studies but also farmers' who were agreeable to CF practices [18,36,63]. More precisely, the aim of the questionnaire was to answer the following research questions:(i) what is farmers' opinion towards contract farming? (ii) Do the socioeconomic characteristics of the farmers matter? and (iii) What are the perceived benefits and drawbacks regarding contract farming? Specifically, the questionnaire consisted of three sections: The first section included closed-type questions on demographic characteristics of the farmers such as gender, age, educational background, and family status. In addition, questions were included, which aimed to describe the economic performance of the farmers such as earnings and total cultivated area. In the second section, farmers were asked about their awareness of the CF concept and their experience and satisfaction regarding CF. Lastly, farmers were asked about their perceptions toward CF, focusing on the perceived benefits and drawbacks from participating in a CF scheme. A copy of the full questionnaire can be obtained from the corresponding author on request.

## 4. Results

### 4.1. Farmers' Socioeconomic Profile

The final study sample comprises 95 males (63.3%) and 55 females (36.7%). Most farmers (62.7%) are not older than 50 years old while 37.3% are aged 51 or over. Regarding their educational level, 64.7% of the farmers have a high school diploma, while 17.3% have completed elementary school. With regards to their experience, the largest proportion of farmers (34.7%) have 26 or more years of experience, while the smallest proportion (10.7%) have 1–5 years of experience. The largest proportion of farmers (33.3%) earn 30,001 euros or more per year, while the smallest proportion (8.7%) earn between 20,001 and 30,000 euros.

As far as the total cultivation area is concerned, the largest proportion of farmers (27.3%) have over 301 hectares of land, while the smallest proportion (13.3%) have 5–50 hectares. More than half of the participants (61.3%) own their land outright, while 32.7% lease their land. Last but not least, nine out of ten farmers (91.3%) stated that they do not belong to an agricultural association or a cooperative.

### 4.2. Attitudes towards CF

The majority of farmers (60.7%) reported being unaware of what CF is. Table 1 presents summarize the statistically significant results of chi-square tests between variables regarding farmers' perceptions of CF and demographic characteristics.Overall, the results of the chi-square tests indicate significant differences in the proportion of individuals who know what CF is based on their gender, years of experience, and total cultivation area. However, there is no significant difference based on their annual agricultural income.

Among those who have joined a CF scheme, 11.3% strongly agreed that they were satisfied, but all of them (29.3%) would recommend participation in a CF program.

**Table 1.** Statistically significant relations resulting from chi-square tests.

| Variables | | $\chi^2$, df, *p*-Value |
|---|---|---|
| Awareness towards CF | gender | $\chi^2 = 13.603$, df = 1, *p* <0.001 |
| -//- | years of experience | $\chi^2 = 10.119$, df = 4, *p* =0.038 |
| -//- | total cultivation area | $\chi^2 = 20.639$, df = 4, *p* <0.001 |
| Satisfaction towards CF | gender | Pearson $\chi^2 = 20.028$, df = 7, *p* <0.01 |
| -//- | years of experience | Pearson $\chi^2 = 30.08$, df = 20, *p* <0.01 |
| Willingness to be informed about CF | annual agricultural income | $\chi^2 = 16.166$, df = 8, *p* =0.040 |
| -//- | total cultivation area | $\chi^2 = 18.619$, df = 5, *p* <0.001 |

The chi-square test shows a statistically significant association between gender and satisfaction level. The percentage of males (62.1%) who are satisfied with the CF program is higher than that of females (48.9%). Accordingly, a statistically significant relationship is found between years of experience and satisfaction level. In particular, the percentage of participants who are satisfied with the CF program they joined is generally higher among farmers with more years of experience. For example, 70.6% of those with 26 or more years of experience are satisfied, compared to a percentage of 33.3% of those with less experience (1–5 years). However, there is no significant association based on annual income or total cultivation area.

Participants were asked to indicate if it would be interesting for them to be informed about CF programs. Data indicated a positive attitude as 68% of the farmers are willing to be informed. The results of the chi-square tests suggest that there is a significant association between farmers' willingness to be informed about CF programs and their annual agricultural income and total cultivation area. However, there is no significant association based on their gender or years of experience.

*4.3. CF Perceptions*

We asked participants to indicate their level of agreement with CF and its potential benefits, challenges, and opportunities, using seven-ordered response levels. Regarding the strengthening of their income through CF, there was a relatively even distribution of responses (30.7% strongly disagree, 31.3% neither agree nor disagree, and 19.3% strongly agree). A total of 26.0% of farmers expressed their strong agreement that CF covers a large part of the cost of agricultural production, followed by neither agreeing nor disagreeing (24.0%) and somewhat agreeing (14.7%). Farmers were then asked whether CF increases the cost of production due to the need to adapt the cultivation to the requirements of the company/processor. Strong agreement reached 26.0%, followed by neither agreeing nor disagreeing (31.3%) and slight agreement (6.0%). Next, farmers were asked whether CF introduces the farmer to new technologies and production methods with 39% of farmers strongly agreeing and 39% strongly disagreeing. As far as the statement that CF supports local communities and development, only 29% of respondents agreed that CF supports local communities and development, with 18.7% strongly agreeing and 8% somewhat agreeing, whereas the majority of respondents (68%) either disagreed or were neutral on this question. The results show that most of the respondents disagreed with the statement that CF promotes sustainable development due to the intensification of cultivation for increased productivity. Specifically, 68% of the respondents either somewhat disagree, slightly disagree, or strongly disagree with the statement. Figure 1 presents the analysis of farmers' responses towards CF perceptions.

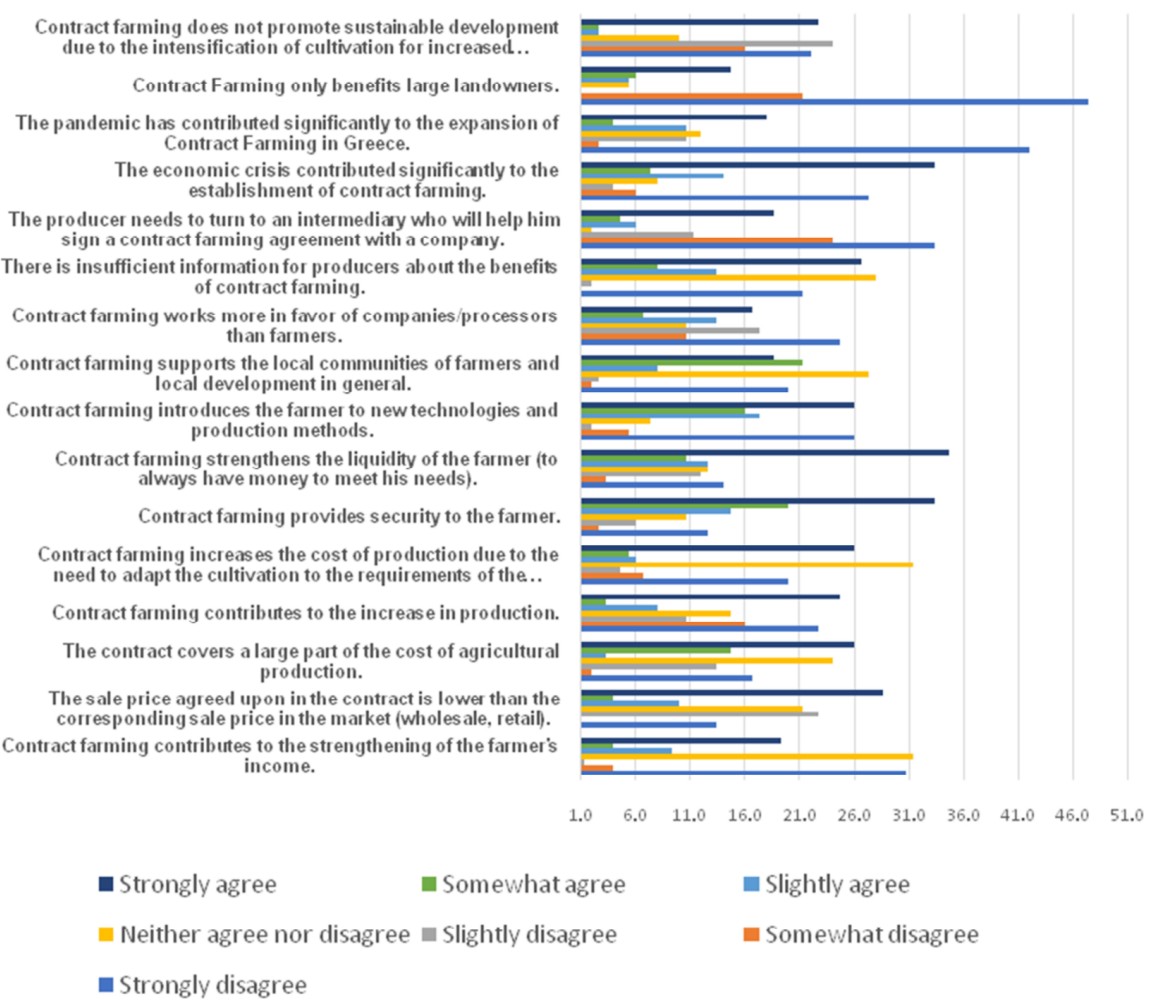

**Figure 1.** CF perceptions (%).

*4.4. Factor Analysis*

Factor analysis identifies "factors" that represent statistically defined constructs in the data regarding farmers' perceptions toward CF. The KMO test measures the sampling adequacy of the data and indicates whether the data is suitable for factor analysis. In our case, the KMO value is 0.866, suggesting that the data are suitable for factor analysis. The Bartlett's test of sphericity resulted in an approximate chi-square value of 1836.115 with 120 degrees of freedom and a significance level of 0.000, indicating that the variables are significantly correlated and suitable for factor analysis. The variable scale has a Cronbach's alpha of 0.802, indicating good internal consistency.

The factor analysis results suggest that the data can be reduced to three factors. Table 2 presents the component matrix that shows the correlations between each statement and each factor. A correlation value close to 1 or −1 indicates a strong relationship between the statement and the factor, while a correlation value close to 0 indicates a weak or no relationship. In this case, three factors were identified.

The first factor, termed "CF Economic benefits", explains 30.992% of the total variance. This factor includes variables that have high positive correlations with statements such as "CF contributes to the strengthening of the farmer's income", "CF covers a large part of the cost of agricultural production", "CF provides security to the farmer", and "CF strengthens the liquidity of the farmer". This factor suggests that CF can provide economic benefits to farmers in terms of income, cost coverage, security, and liquidity.

**Table 2.** Component matrix.

| Statements | Components | | |
|---|---|---|---|
| | **1** | **2** | **3** |
| **"CF Economic benefits"** | | | |
| CF contributes to the strengthening of the farmer's income. | 0.712 | −0.383 | 0.159 |
| The sale price agreed upon in the contract is lower than the corresponding sale price in the market (wholesale, retail). | | −0.266 | |
| CF covers a large part of the cost of agricultural production. | 0.815 | 0.245 | 0.114 |
| CF contributes to the increase in production. | 0.828 | | −0.362 |
| CF increases the cost of production due to the need to adapt the cultivation to the requirements of the company/processor (e.g., modification of cultivation for specific quality). | 0.550 | 0.190 | −0.502 |
| CF provides security to the farmer. | 0.835 | −0.110 | −0.280 |
| CF strengthens the liquidity of the farmer. | 0.763 | 0.137 | |
| **"CF challenges and risks"** | | | |
| CF introduces the farmer to new technologies and production methods. | 0.690 | −0.553 | 0.217 |
| CF supports the local communities of farmers and local development in general. | 0.709 | −0.481 | 0.346 |
| CF works more in favor of companies/processors than farmers. | −0.150 | 0.337 | 0.673 |
| CF does not promote sustainable development due to the intensification of cultivation for increased productivity (more fertilizers, more working hours, etc.). | 0.410 | 0.303 | −0.347 |
| There is insufficient information for producers about the benefits of contract farming. | 0.119 | −0.233 | 0.513 |
| The producer needs to turn to an intermediary who will help him sign a contract farming agreement with a company. | 0.342 | 0.739 | |
| CF only benefits large landowners. | 0.122 | 0.837 | |
| **"External factors"** | | | |
| The economic crisis contributed significantly to the establishment of CF. | 0.403 | 0.594 | 0.498 |
| The pandemic has contributed significantly to the expansion of CF in Greece. | 0.203 | 0.124 | 0.836 |

The second factor, termed "CF challenges and risks", explains 17.229% of the total variance. This factor includes variables that have high positive correlations with statements such as "CF introduces the farmer to new technologies and production methods", "CF supports the local communities of farmers and local development in general", and "CF does not promote sustainable development due to the intensification of cultivation for increased productivity", and can be interpreted as the "Social, Technological and Environmental Impact" factor. This factor suggests that CF can have social, technological, and environmental effects, both positive and negative, such as introducing new technologies, supporting local communities, and intensifying cultivation practices.

The third factor, termed "Sustainable production", explains 15.337% of the total variance. This factor includes variables that have high positive correlations with statements such as "The pandemic has contributed significantly to the expansion of CF in Greece" and "The economic crisis contributed significantly to the establishment of CF", and can be interpreted as the "External Factors" factor. This factor suggests that external factors such as the economic crisis and the pandemic can influence the establishment and expansion of CF.

Overall, the results of the factor analysis suggest that there are both potential benefits and risks associated with CF, and that contextual factors play an important role in shaping its adoption and implementation.

## 5. Discussion

Much attention has been paid over the last decades to CF in both developed and developing countries [41,64,65]. Involvement in various CF schemes presents different degrees of opportunities and constraints [16], and, thus, perceptions of farmers regarding CF are of critical importance for its adoption.

Most respondents in our research are men above 35 years of age with considerable experience in farming. Most of them have graduated from secondary school, not pursuing further studies, and about 1 in 10 has a university degree. Despite female farmers' role in rural economies [66], their percentage in our study seems quite limited. This is a rather expected finding, considering that "local cultural standards and values affect the overall integration of women in agriculture despite all incentives and contemporary policies" [67]. The demographic and economic variables such as gender, years of experience, and income are of unequivocal importance for the Greek government and the institutions concerned to investigate how Greek farmers assess opportunities, challenges, and risks related to CF to provide the framework needed to facilitate the transition to modern and sustainable agriculture. Agreement exists in the literature that the socioeconomic profile has a decisive impact on joining a CF scheme [32,33,63,68] (without a consensus on both the sign and significance of variables such as sex, age, and education on the probability of participation [15]. Therefore, researchers and policymakers interested in the development of CF should be wary of the heterogeneity for the development and success of policies related to CF as also indicated by Wang et al. [15].

The research findings also indicate a low level of CF awareness among Greek farmers. This is a rather expected finding provided that in Greece agriculture remains an economic activity where tradition plays a significant part and practices flow from generation to generation [14], in combination with structural problems of the sector such as the lack of a skilled workforce [4]. As also indicated by previous research in Greece, CF has been informally in place for years, but it still remains limited in terms of both variety of crops and cultivated area [14]. Eaton and Shepherd argue that CF opens up new markets [1]. Chaniotakis pinpoints that the Greek agrifood sector has to adjust its productive model towards the market for high-quality products [69]. CF is strongly linked to better product quality [10,48,70] and, thus, should be further promoted by the Greek State.

Focusing on agriculture as a field of innovation both in technical as well as financial terms, introduction of the CF concept can be regarded as a case of diffusion of innovation [71], even if the contracting of crops was already widespread in ancient Greece [1]. The above finding should be seriously considered by the Greek government and the institutions concerned in order to raise awareness of CF concept. In this context, representatives of the public sector at national, sub-national, and local level may develop educational programs to increase theoretical and practical understanding of CF, providing both insights and tools for farmers to become acquainted with CF and incentives to actively engage to a CF scheme, if interested. In line with Chaniotakis [69], as innovation in the agri-food value chain can be initiated from nontraditional direct sector participants, such as banks, innovation in CF finance based on trust and long-term cooperation should be also further elaborated.

Furthermore, factor analysis revealed three factors, namely, "CF economic benefits", "CF challenges and risks", and "sustainable production". With regard to the CF economic benefits, they are widely recognized by researchers in both developed and developing countries [45,72]. Risk reduction and transaction cost savings are predominant for farmers in developed countries [15], while an increase in income, a reduction in poverty, and an improvement in the livelihoods of farm households are indicated in the developing countries [16,53]. It is worth mentioning that, in the case of Greece, 17,000 people are estimated to have entered into the agriculture sector in 2011 as a consequence of the

economic crisis that led people to go back-to-the-land [73], that is, after all explained by a pre-existing connection to land and rural areas of Greeks and its "rediscovery" motivated by economic reasons [74]. However, dissatisfaction with contract schemes in spite of economic benefits is also noted [18].

Given that CF has a significant effect on advancing productivity [30], challenges and risks are emerging inextricably linked to new technologies, vertical integration, and ongoing complexity. Asano-Tamanoi [7] notes that "farmers today stand in relations of growing complexity with various "others" for the purpose of agricultural production". Rehber [75] suggests that "CF is not a panacea to solve all related problems of agricultural production and marketing systems". However, in this challenging environment, the development of training programs by academic institutions and professional associations is even more urgent for Greek farmers so as to cope with the fourth industrial revolution (Industry 4.0) and its emerging technologies such as the Internet of Things (IoT), robotics, Big Data, Artificial Intelligence (AI), and blockchain technology as discussed by Liu et al. [76]. Participation in international conferences may also offer significant learning opportunities to farmersfor obtaining a CF mindset.

Furthermore, external factors such as an economic crisis are associated with CF sustainable production. Given that agricultural environmentally sustainable production plays a dominant role in the future evolution of agriculture [77], CF should be a strategic tool for smart, sustainable, and inclusive growth. In this context, VabiVamuloh et al. [78] argue that "CF can help achieve Sustainable Development Goals (SDGs)". Recently, pressure on farming from the pandemic has brought a rapid evolution of new ways of cooperation between farmers themselves and other market players [79]. It has also been suggested [80] that wider adoption of CF will be one of the long-term outcomes of the pandemic, provided it creates a sense of security and flexibility in farmers, so that they will better focus on production. However, a legal system and legislation to support farmers involved in CF is needed. Last but not least, researchers [19] suggest that, despite its problems, CF seems to have a bright future in the post-COVID era.

## 6. Conclusions

Despite the rise and expansion of CF schemes both in the developed and the developing world, little empirical research exists on CF practices in Greece. As Greek people are going back-to-the land, CF is emerging as a concept of crucial importance for the domestic agricultural sector, especially in the post-COVID era. Within the circumstances of the global COVID-19 pandemic, farming was under multiple pressures, ranging from the physical well-being of farmers themselves to lockdowns that affected production. In this context, the current research aimed to explore Greek farmers' perceptions towards CF in the post-COVID era. It attempted to explore if farmers are aware of the CF concept, how they perceive it, and how they assess the potential benefits and challenges linked to CF.

We suggest that Greek farmers are not involved to a great extent in a CF scheme due to the low level of CF awareness. Furthermore, the findings indicate that they do understand economic benefits and social, technological, and environmental challenges and risks associated with CF. Given that the Greek agricultural sector is unique "due to its structure, history, product categories involved, processes, legal environment and participants' mentality" [69], policymakers and researchers need to pay attention to the CF concept and its implementation in Greece, promoting sustainable development.

Our study is not without limitations. Its findings may not be adequately generalizable in other contexts, since the study relies on self-reported data, and also the sample was restricted to a specific region of Greece. This study also does not distinguish between different stakeholders, i.e., agri-food industry managers, banks, and agricultural policy makers.

Firstly, future studies may be conducted for different regions of the country and may include all CF stakeholders involved. Therefore, a comparative study between farmers from all regions of the country and other stakeholders may shed light on the CF concept. Research would benefit from the analysis of data on consumers' beliefs and attitudes

towards CF practices. Moreover, further analysis of data sets from other countries would help develop the existing literature, leading to potentially more generalizable outputs.

**Author Contributions:** Conceptualization, G.M., E.S. and M.B.; formal analysis, G.M., E.S. and M.B.; investigation, G.M., E.S. and M.B.; methodology, G.M., E.S. and M.B.; resources, G.M., E.S. and M.B.; software, G.M., E.S. and M.B.; supervision, G.M., E.S. and M.B.; validation, G.M., E.S. and M.B.; visualization, G.M., E.S. and M.B.; writing—original draft, G.M., E.S. and M.B.; writing—review and editing, G.M., E.S. and M.B. All authors have read and agreed to the published version of the manuscript.

**Funding:** This research received no external funding.

**Institutional Review Board Statement:** Not applicable.

**Informed Consent Statement:** Informed consent was obtained from all subjects involved in the study.

**Data Availability Statement:** The data presented in this study are available on request from thecorresponding author. The data are not publicly available due to confidentiality requirements.

**Acknowledgments:** The authors wish to thank the editor and four anonymous reviewers for their very useful and constructive comments. All errors and deficiencies are the responsibility of the authors.

**Conflicts of Interest:** The authors declare no conflict of interest.

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
