# Peer review of "Farmers’ Perception on Contract Farming in the Post-COVID Era: Empirical Study in Greece"

_sustainability, doi:10.3390/su152014696_

Round 1
Reviewer 1 Report
General comment:
Overall, the paper is quite interesting and well written. It is interesting to see research on the specific area of Contract farming as there is limited research in this whole area of collaborative farming. The paper highlights some interesting results and has the potential to provide a valuable contribution to this under explored research area.
Specific Section Comments:
Abstract:
· In the abstract the objective of the paper could be more clearly outlined. For example, the abstract states that “This paper focuses on Greek farmers’ perceptions toward CF”. This appears too brief.
Methodological framework:
· The methodological framework section is too brief. More detail is required.
· There is no detail on how the data was analysed, this is required.
· Also, perhaps include if the data analysis method has been used in prior studies on this topic or area of research and supply references.
Results:
· Overall the results are interesting, however, in some areas the results could be more clearly presented.
· In section 4.3, for the CF perceptions there appear to be 7 questions asked and the authors describe the relative response percentages of each statement, which seems quite repetitive. Perhaps the 7 statements and the associated percentage response rates could be presented in a table and then followed by some key observations or discussion.
· In Section 4.4, in the discussion for Table 1 the authors describe how statements relate to three factors “CF Economic Benefits”, “CF Challenges and Risks” and “Sustainable Production”. Perhaps in Table 1 these three factors could be included to label and group together those statements that relate to each factor.
Discussion and Conclusion
· The final paragraph should be developed to more comprehensively outline the limitations of the study and to highlight a number of areas of future research.
· This section could more comprehensively address the potential policy and farm management implications of the findings of this study.

Author Response
Dear Reviewer,
We would like to thank you for your insightful comments and efforts towards improving this manuscript. In the following, we highlight our effort to address your concerns for amendments. Please see below a point-to-point list of changes.
On behalf of the authoring group
George Malindretos

Reviewer 2 Report
Dear author,
The manuscript presents an interesting and promising study on farmers' perception on contract farming. However, due to a number of critical drawbacks of the current version of the manuscript, I can not recommend the submission for acceptance. Instead, I would suggest the following revisions to be made prior the manuscript could be reconsidered for the publication in Sustainability.
The Abstract should be reworked. The author should start with defining the problem and demonstrating the overall relevance of the issue and the specific relevance of the issue for Greece. Then proceed with gaps and how the study addresses them. Then methodology and findings. Then contributions to the literature and practical implications of the findings. The Abstract should be as brief and focused as possible, but all the above issues should be emphasized.
In the Introduction, the author should elaborate the gaps in studies and policies related to Greece. The specific relevance of the study for the country remains unclear. No specific gaps have been revealed, especially those related to sustainable development (the scope of the journal) and the effects of the pandemic on the agricultural sector (as stated in the paper title).
The Methodology section must be expanded. In its current form, the presentation of the methodology is unacceptable. The author must explain the selection of the methods (in a critical manner), the construction of the questionnaire (why those particular questions and variables), and the construction of the array of respondents (which respondents, how they are accessed, selected, etc., what about the representativeness of the array). The sample questionnaire must be provided in the appendix.
The discussion section should be divided from the conclusion. The author should clearly demonstrate the novelty of the findings and the similarities/differences of the author's findings with/from other studies in the area.
The quality of the English language and style should be improved
Author Response

(The authors gave the same response as above.)

Reviewer 3 Report
This is an interesting paper which focuses on farmers' perceptions on contract farming in Greece.
However more details on the research period, the questionnaire structure and the methods are needed.
General speaking, the paper was easy to be followed and the language encourages reading the text.
Author Response

(The authors gave the same response as above.)

Reviewer 4 Report
Manuscript number: sustainability-2542746
Title: Farmers’ perception on Contract Farming in the post-Covid era. Empirical study in Greece
Comments for the Author
This article covers the important concept of contract farming. Contract farming is getting popularized with population explosion and globalization. Marginal farmer is in danger and their profitability is a challenge. This scheme, contract farming is an important agricultural arrangement for the small/marginal farmer that involves formal agreements between farmers and agricultural industries. This practice has increased importance due to its potential to bring numerous benefits to both the agribusiness industry and farmers, as well as the broader economy. As the authors have covered an important topic I recommend this manuscript for further consideration in the journal with some suggestions given below.
Some minor suggestion
· Abstract: The quantitive finding is lacking (like % of farmer awareness about CF farming, and so on) suggested to add qualitative results.
· Line no 93: Add relevant references to, and other countries (???; ???; ???).
· Line no 186: Overall, the final study sample consisted of 150 questionnaires suitable for analysis, justify how a sample size of 150 was selected???
· Line no 198: typographical error (add % sign after 63.3).
· Suggested splitting of discussion and conclusion section (Discussion section can be merged with result section, if authors agree).
· Novelty of the study: Improve novelty in the introduction section.
NA
Author Response

(The authors gave the same response as above.)

Round 2
Reviewer 2 Report
The paper has been improved substantially compared to the Round 1 version. The author has adequately and sufficiently addressed my recommendations and comments. The paper can be considered for acceptance after thorough proofreading - the quality of the English language and style could be improved
The paper can be considered for acceptance after thorough proofreading - the quality of the English language and style could be improved